# The utility of infectious disease modelling in informing decisions for outbreak response: A scoping review

Duaa Rao[1,2☉], Aleena Tanveer[1,2☉], Emil Nafis Iftekhar[1☉*], Sophie Alice Müller[1], Katharine Sherratt[3], Klara Röbl[1], Paola Carrillo-Bustamante[1], Katharina Heldt[4], Julia Fitzner[5], Johanna Hanefeld[1,6], Sebastian Funk[1,3]

**1** Centre for International Health Protection, Robert Koch Institute, Berlin, Germany, **2** Institute of International Health, Charité—Universitätsmedizin Berlin, Berlin, Germany, **3** Centre for Mathematical Modelling of Infectious Diseases, London School of Hygiene & Tropical Medicine, London, United Kingdom, **4** Method Development, Research Infrastructure and Information Technology, Robert Koch Institute, Berlin, Germany, **5** World Health Organization Hub for Pandemic and Epidemic Intelligence, Berlin, Germany, **6** Department of Global Health and Development, London School of Hygiene & Tropical Medicine, London, United Kingdom

☉ These authors contributed equally to this work.
* IftekharE@rki.de

## Abstract

Infectious disease modelling plays a critical role in guiding decisions during outbreaks. However, ongoing debates over the utility of these models highlight the need for a deeper understanding of their exact role in decision-making. In this scoping review we sought to fill this gap, focusing on challenges and facilitators of translating modelling insights into actionable policies. We searched the Ovid database to identify modelling studies that included an assessment of utility in informing policy and decision-making from January 2019 onwards. We further identified studies based on expert judgement. Results were analysed descriptively. The study was registered on the Open Science Framework platform. Out of 4007 screened and 12 additionally suggested studies, a total of 33 studies were selected for our review. None of the included articles provided objective assessments of utility but rather reflected subjectively on modelling efforts and highlighted individual key aspects for utility. 27 of the included articles considered the COVID-19 pandemic and 25 of the articles were from high-income countries. Most modelling efforts aimed to forecast outbreaks and evaluate mitigation strategies. Participatory stakeholder engagement and collaboration between academia, policy, and non-governmental organizations were identified as key facilitators of the modelling-for-decisions pathway. However, barriers such as data inconsistencies and quality, uncoordinated decision-making, limited funding and misinterpretation of uncertainties hindered effective use of modelling in decision-making. While our review identifies crucial facilitators and barriers for the modelling-for-decisions pathway, the lack of rigorous assessments of the utility of

**Data availability statement:** The extracted data from the included publications is attached as a Supporting information.

**Funding:** DR, AT, ENI, SM, PCB, KH, KR, JH, SF are supported by the Robert Koch Institute. DR and AT are in part funded, and ENI and KR are fully funded by the World Health Organization. KS and SF are supported by the Wellcome Trust (210758/Z/18/Z). JF is supported by the World Health Organization. The external funder Wellcome Trust had no role in study design, data collection and analysis, decision to publish, or preparation of the manuscript.

**Competing interests:** The authors have declared that no competing interests exist.

modelling for decisions highlights the need to systematically evaluate the impact of infectious disease modelling on decisions in future.

## Introduction

### Context: Infectious disease modelling as a tool for outbreak response

Policymakers in public health face complex decisions during infectious disease outbreaks, such as AIDS, Ebola, mpox and COVID-19. These decisions often involve balancing multiple competing priorities under significant uncertainty, including the timing and scale of interventions, resource allocation, and the social and economic impacts of control measures (e.g., [1]). This complexity underscores the critical need for effective decision-making tools. Mathematical modelling has emerged as an important tool in this context, often referred to as 'infectious disease modelling' (e.g., [2,3]). It supports outbreak response in multiple ways: Modellers can for example estimate key epidemiological parameters, forecast outbreak size and assess the potential impact of different control measures (e.g., [4,5]). They do so by describing epidemiological data and phenomena by equations or explicit rules, and performing simulations or statistical analyses based on them. Thereby, modelling translates complex epidemiological data into actionable insights for decision-makers as highlighted by its use during outbreaks of, e.g., A/H1N1 influenza, Ebola and COVID-19 (e.g., [4,5]).

### Context: Challenges in the use of infectious disease modelling for outbreak response

However, there is extensive debate about whether these models could be trusted and used effectively to guide policy choices (e.g., [1]). Especially when there are conflicting results, models face criticism from politicians, the press, experts from other disciplines and the public (e.g., [5]). These tensions sometimes cause potentially avoidable delays in implementing measures and makes modellers apprehensive to further contributing to outbreak response (e.g., [6]). Hence, it can be challenging to utilise modelling successfully for outbreak response.

### Context: Need for a better understanding of the use of infectious disease modelling for outbreak response

These challenges highlight the need for continued efforts in enhancing how the results of infectious disease modelling for outbreak response are used, as recognised by the 'Lancet Commission on Strengthening the Use of Epidemiological Modeling of Emerging and Pandemic Infectious Diseases' [7].

### Rationale: Need for a scoping review

Previously, a scoping review on evidence use for decision-making during infectious disease outbreaks was conducted that tangentially considers the role of modelling evidence [8]. However, as it lacks a detailed perspective on modelling, focuses on

Europe and considers records only from before the COVID-19 pandemic, an updated more comprehensive overview is needed.

To improve the utility of modelling for decisions, it is necessary to identify typical characteristics, facilitators and challenges of the modelling-for-decisions pathway. Such mapping is particularly timely given the ongoing need to strengthen the science-policy interface in outbreak response, such as for the ongoing mpox epidemic.

At the time of writing, there are two other projects similar to ours: First, there is a pre-registered scoping review focused on the question 'What defined policy-relevant advanced analytics over time during the COVID-19 pandemic in Europe?' [9]. Second, there is a study that was published after we had already conducted and pre-registered our search and screening; it also contains a scoping review focusing on the effective translation of 'modelled evidence into policy decisions' in Lower- and Middle-Income Countries in the context of the COVID-19 pandemic [10]. We complement this work by conducting a scoping review that is not limited geographically and also considers diseases beyond COVID-19.

### Questions and objectives: What has been the utility of infectious disease modelling in informing decisions for outbreak response?

This review examines how mathematical modelling has informed decisions during infectious disease outbreaks and epidemics. We intended to investigate what types of infectious disease models have been used and how they have been utilised in decision-making, with particular focus on evaluations of how useful models were in informing disease outbreak control or mitigation efforts.

Our analysis addresses several interconnected questions about modelling-for-decisions translation: What characterizes modelling-for-decisions pathways? What are the facilitators and barriers for successful integration of modelling evidence into decisions? We examine publications that reflect on or evaluate modelling engagement with decision-makers for outbreak response, including who has studied these questions, in what contexts, and what recommendations emerge for improving modelling-for-decisions pathways.

## Methods

A review protocol for this study has previously been published with the Open Science Framework on June 5 2024, with updates on June 6 2024, July 17 2024 and 20 February 2025, and can be accessed at https://osf.io/cd4qa/ [11]. The writing of this manuscript followed the PRISMA guidelines for Scoping Reviews (PRISMA-ScR) [12]. The PRISMA-ScR checklist can be found in S1 PRISMA Checklist.

### Eligibility criteria

The eligibility criteria are based on the PCC framework (population, concept and context), as suggested by the Joanna Briggs Institute [13]. The detailed inclusion and exclusion criteria by categories as well as types of evidence sources are presented in Table 1.

### Search strategy

To define the search components, we adapted the CoCoPop scheme (Condition, context, population) to CoCoPop + E to include the evaluation component [14]. We searched the database Ovid (Medline, Embase) on June 5 2024 using the search strategy depicted in Fig A in S1 File. The rationales for the details of the search are listed in Table A in S1 File.

Following the search, all identified citations were collected and uploaded into EndNote X7 (Clarivate Analytics, PA, USA) and duplicates were removed. Titles and abstracts were screened by the research team for assessment against the eligibility criteria for the review. Additional literature was added through expert judgement in two ways: (a) The senior

**Table 1. Eligibility criteria for publications in the review and the rationales for selecting these criteria.**

| Category | Criterion (inclusion: white, exclusion: orange) | Rationale |
|---|---|---|
| **Population** | Models on humans | We are interested in outbreaks in human populations. |
| **Concept** | Models to inform disease outbreak control or mitigation efforts (including policy making regarding resource management such as hospital management) | We are interested in the role of models in informing policies for outbreak response. Outbreak response entails control and mitigation efforts. Resource management falls under these efforts. |
| | Publications that reflect on or evaluate their engagement with policies for outbreak response. | We are interested in the role of models in informing policies for outbreak response. |
| | Models to inform pharmaceutical interventions | The research body on these models is too big for the scope of this review. |
| | Models on any specific non-pharmaceutical interventions during the COVID-19 pandemic | There are multiple already ongoing and performed reviews and a large evidence base. |
| | Models with only theoretical link to policy making | We are interested in the role of models in *actually* informing policies for outbreak response. |
| | New model generation or comparative models | |
| **Context** | Models in response to acute infectious disease outbreaks | We are interested in the role of models in informing policies for acute infectious disease outbreak response. |
| | Models on pathogens SARS-CoV-2, measles, Zika, C. diphtheriae, mpox, HIV, TB, influenza and Ebola | To keep the scope manageable, we focus on some key pathogens only. |
| | Models on cost-effectiveness | The research body on these models is too big for the scope of this review. |
| **Type of evidence sources** | Publications: peer reviewed | Peer-review indicates a basic quality control of the considered publications. |
| | All type of studies | We are interested in any type of study that provides information about the role of models in informing policies for outbreak response. |
| | Languages: English | As our working language was English, we chose to include only English publications in the search. |
| | Time: last 5 years (January 2019 - May 2024) | We are interested in current practice. Starting in 2019 also allows us to include the vast literature related to the COVID-19 pandemic. |
| | Reviews | We want to understand the specific modelling and policy context for each case study. |

The cells in orange contain exclusion criteria, the cells in white contain inclusion criteria.

authors of this review suggested relevant additional publications and (b) the selected literature of a similar scoping review [10] was considered. These additional publications were also checked against the eligibility criteria and included/excluded accordingly.

## Article selection

The full text of selected citations was assessed in detail against the eligibility criteria by the research team. Two reviewers, AT and DR, independently screened all articles for inclusion. Disagreements were resolved by additional reviewers.

## Data extraction

A data extraction table (via Google Sheets) was developed and updated in an iterative way, resulting in five broad and 21 sub categories including their definitions (Table 2). The table updates comprised clarifications of definitions as well as re-conceptionalisations and re-clustering of categories to achieve better coherence.

**Table 2. Broad and sub categories for the data extraction along with their definitions.**

| Broad category | Sub category | Definition |
|---|---|---|
| publication details | author | First author's name |
| | author affiliation | Authors' affiliations by type: academic or policy organisation, or dual (academic/policy) affiliation |
| | publication year | Year the article was published |
| | title | Title of article |
| | journal | Name of journal in which article was published |
| | DOI | DOI of article |
| study design | study period | Duration of the study |
| | geographical setting | Geographical location considered in the model. |
| | target population | Population for/about which the disease modelling was done |
| | disease | Name of the disease or pathogen considered in the model |
| policy decision | initiative | Authority/organisation that started the modelling initiative |
| | policy objective | All recommendations by modellers in the study, regardless whether any of them were implemented as a policy |
| | decision maker | Authority implementing the model recommendations or authority addressed |
| | approach for policy interaction | Communication between modellers, policy makers or any other professionals/organisations involved including timing/frequency and format of communication |
| modelling | type of model | Type of model used for analysis, e.g., compartmental or agent-based model |
| | stakeholder input | Stakeholder input used in the quantitative modelling process, e.g., in selecting variables or setting priors |
| reflections on model utility for policy | role of modelling | Role that modelling played in the policy making process, e.g., providing forecasts, assessing impact of interventions |
| | validity | Quantitative or qualitative evaluation of model projections, e.g., accuracy against observed data or evaluation against another model |
| | barriers | Problems faced during the process of modelling, e.g., missing data |
| | enablers | Collaborations or sources that facilitated the modelling process |

Data from each included publication was extracted independently by two reviewers, respectively. The involved reviewers were AT, DR, and ENI. In case of immediate agreement, the wording of the first reviewer was chosen. Complementary data was added, where necessary. Any arising disagreements were discussed and resolved between the two reviewers. Where the data extraction for certain publications was deemed difficult by the reviewers, help was sought from additional co-authors.

### Synthesis of results

The purpose of this scoping review is to map the research done on the utility of infectious disease modelling in informing decisions for outbreak response, as well as to identify any existing gaps in knowledge on this topic. As we did not find objective assessments of utility to include into our review but subjective reflections by practitioners, we instead summarised aspects of the modelling-for-decisions pathway that are claimed to be essential for the utility by the authors of the included publication. We performed a descriptive qualitative content analysis and synthesis of the data extraction results along the categories that we chose in our data extraction table.

## Results

### Article selection

Out of the 4007 articles identified through database search via 3 databases 3965 were screened for inclusion after removal of duplicates (Fig 1). Out of the title and abstract screened articles, the full texts of 42 articles were screened

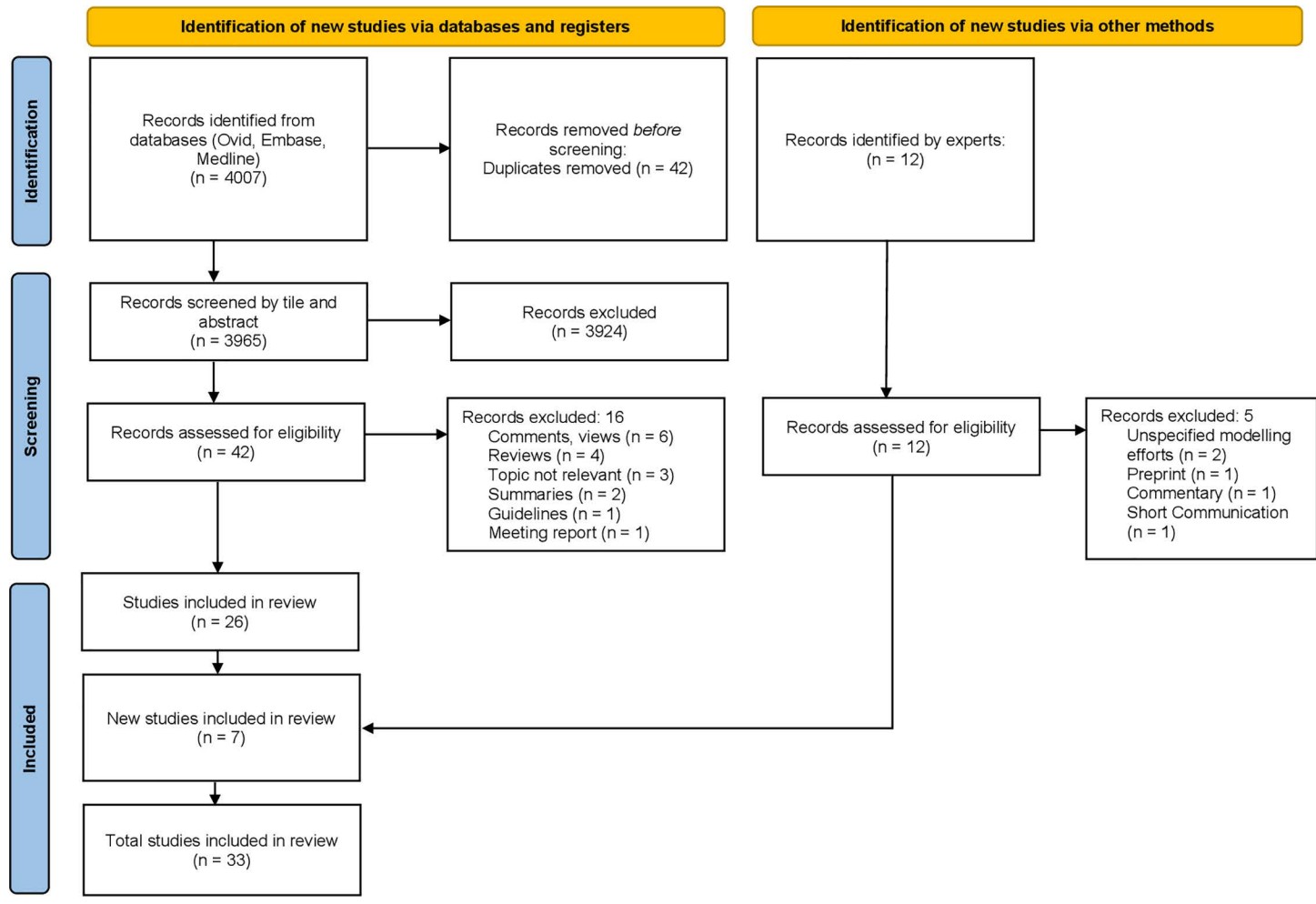

**Fig 1. Flow diagram of the article selection process.**

against the eligibility criteria and 26 of them were included in the review. Full texts of additional 12 articles suggested based on expert judgement were also screened against the criteria and 7 of them were included in the review. Data was extracted from all 33 included articles (S1 Data).

## Summary of extracted data

The full extracted data are presented in the data extraction matrix (S1 Data).

No study provided objective assessments of utility. 31 of the 33 included articles are descriptions and reflections by modellers about their own modelling efforts where they highlight aspects they deemed relevant for the utility of their work in informing decisions for outbreak response. Only in two articles, the perspectives of decision-makers are portrayed [15,16]. In the following, we summarise these subjective reflections.

The summary of the extracted data is presented along the categories of the data extraction. No clear relationships or patterns between the different categories could be identified, e.g., whether a specific modelling approach was more or less prone to a certain barrier. Hence, the categories are summarized independently.

The included literature represents all continents (Fig 2). However, most of the studies come from High Income Countries (HICs), with multiple articles on modelling efforts in the USA (13 articles) and UK (5 articles). Most Low-and-Middle Income Countries (LMICs) that are highlighted in the graph are considered through the included articles related to the same modelling consortium ('CoMo Consortium') [17,18]. These articles were included based on the study by Owek et al. [10]. While the CoMo Consortium informed decision makers in LMICs, most modellers in the consortium were from HICs.

Out of the 33 articles included, 27 had models on COVID-19, one combined modelling of COVID-19 and Influenza [19] with two models on Influenza alone [20,21], one on Diphtheria [22], one on HIV [23] and one article considered models on various infectious diseases including Influenza, Pertussis, Tuberculosis and others [24]. Our data extraction yielded no key differences between modelling efforts about COVID-19 and other diseases.

Visualisations of the considered diseases, publication years and considered population groups can be found in the supplementary information (Figs B, C, and D in S1 File).

**Authors' affiliations and modelling initiatives.** Out of the 33 included articles, 17 articles were by authors with solely academic affiliations while the remaining 16 articles were authored by both those with academic and policy affiliations.

Infectious disease modelling efforts were initiated at various levels (Table 3). While most modelling efforts were conceptualised and started either by scientists working in academia or by officials at decision-making authorities, some initiatives were also seen as a collaborative work between academia and local, regional or national decision-making authorities. All modelling initiatives were found to be in line with the modelling infrastructures defined in a previous global analysis on the use of epidemiological modelling in health crises, i.e., "one modelling team; multiple small teams functioning as one; a consortium consisting of multiple teams and multiple models led by a modelling committee; and finally, teams working in isolation feeding independently into government" [2].

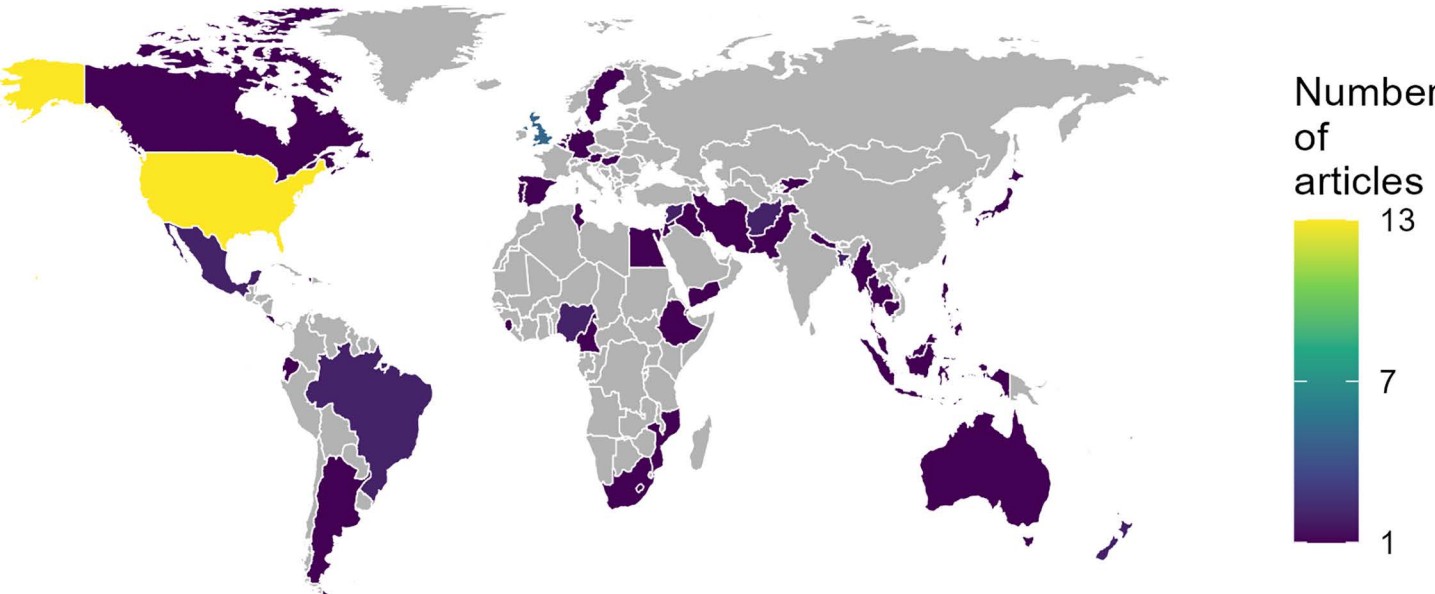

**Fig 2. While many countries worldwide are technically represented, most of the experiences described in this review involve High Income Countries (HICs).** World map highlighting the countries that were considered in the included articles. Although all continents are represented, the majority of articles consider HICs. Note that some articles consider multiple countries; two of these articles account for most of the countries of the Low-and-Middle Income Countries (LMICs) in this figure [17,18]. Both refer to a modelling consortium of mainly HIC modellers informing LMIC decision makers.

**Table 3. Disease modelling efforts were initiated by various actors.**

| Modelling initiatives | Example(s) |
|---|---|
| Academia | • Scientists working pro bono for national COVID-19 response, Israel [25]<br>• CoMo Consortium, LMICs [18] |
| Collaboration between academia and policy | • Collaboration between the Ministry of Health, Pan American Health Organisation and scientists from academia; Costa Rica [26]<br>• South African COVID-19 Modelling Consortium, a collaborative effort between academic and government structures [27] |
| Policy and decision-making authority | • Austrian governmental crisis unit, Ministry of Health [28];<br>• UK Health Security Agency [20];<br>• US Centers for Disease Control [21];<br>• University of Miami Hospitals and Clinics, USA [29] |
| Non-governmental organisation | • Global Challenges Research Fund Project [22] |
| International governmental organisation | • European Centre for Disease Prevention and Control [30] |

The initiators were policy actors and decision-makers as well as modellers themselves.

**Policy objectives and role of modelling.** Infectious disease modelling was done to cater various policy objectives. Most modelling efforts were done to guide operational planning and decision-making in the context of high uncertainty. Mathematical models were used in understanding transmission dynamics, and to forecast the scale of disease outbreak by providing both short- and long-term projections of confirmed cases, hospitalizations, and deaths, while aiding in healthcare resource management and allocation such as hospital bed occupancy and staffing needs [16,19,21,25,27,31–33].

Many modelling approaches were also undertaken to assess the effects and impacts of various mitigation strategies and answer policy questions under different disease outbreak scenarios. For instance, during the COVID-19 pandemic, modelling determined the timeframe of the lifting of the state of emergency (SOE) in Tokyo while controlling infection and minimizing the economic loss [34], while the CoMo Consortium's model was used to determine lockdown and school opening and closing strategies [17]. In New York, the modelling aimed to meet the benchmark of the 'HIV Ending the Epidemic Initiative' under the most realistic conditions, with a focus on prioritizing men who have sex with men [23].

**Modelling approach.** In 15 of the modelling efforts, practitioners developed and used compartmental models, i.e., models based on differential equations. 14 of the articles used Agent Based Models (e.g., [28]) or stochastic/statistical models (e.g., [35]). 10 studies mentioned more statistical approaches, such as using neural networks or Generalized Additive Models (GAM) (e.g., [20,30,33,36]). 4 studies utilized ensemble approaches, synthesizing results from multiple models that included Agent based, Stochastic and compartmental models among others - often by different modelling teams (e.g., [21]). Note that many articles considered multiple models and model types.

**Decision maker.** Infectious disease modelling was used to inform decision-makers across multiple governance levels. (Table 4). While some articles considered decision makers on one governance level, some considered multiple levels, e.g., in the Philippines [37] and South Africa [27].

**Approaches to policy interaction.** We identified various communication formats used by modellers to convey their findings to policymakers, other modellers, and other relevant professionals and organizations.

These communication approaches ranged from formalized, structured processes to more ad-hoc arrangements. Formal structures included established reporting templates [28], regular meeting schedules (e.g., [23]), and dedicated communication channels (e.g., [18]). Diverse communication methods were employed for these interactions, from daily reports (e.g., [29]) to weekly communications (e.g., [39]) and monthly meetings (e.g., [15]). The formats included written reports (e.g., [19]), face-to-face or virtual meetings (e.g., [40]), and formal presentations (e.g., [17]). Several initiatives established

**Table 4. Infectious disease modelling was used to inform decision-makers across multiple governance levels.**

| Governance level | Example(s) |
|---|---|
| International governmental organisation | European Centre for Disease Prevention and Control [30] |
| National authority | Austrian governmental crisis unit, Ministry of Health [28]; Japan's Cabinet Office, Prime Minister's Office, Advisory Board of Ministry of Health, Labour and Welfare [34]; US Centers for Disease Control [21,33] |
| State/regional authority | Pará State Department of Public Health, Brazil [32] |
| Local decision maker | Local health authority Berlin-Reinickendorf, Germany [35] |
| Non-governmental organisation | Médecins Sans Frontières [22] |
| Hospital | Local National Health System and hospital managers, UK [38] |

Decision-makers who used modelling to inform their decisions were not only from political institutions but also from, e.g., hospitals.

dedicated coordination teams or workstreams to manage the interface between modellers and decision-makers (e.g., [41]).

Hierarchical paths were frequently observed, where results were first presented to technical groups, then to advisory bodies, and finally to government through chief scientific advisors (e.g., [42]).

A notable theme across multiple studies was the emphasis on participatory and collaborative approaches to ensure effective knowledge translation. This was highlighted in The CoMo Consortium's stepwise method to establish channels of communication between local decision-makers and modellers, efforts by Médecins Sans Frontières and Canada's bi-directional communication between modellers and decision-makers via multiple workshops to identify translational barriers. [18,22,24]. The adaptation of communication methods to different audiences was also common, with both technical reports and simplified briefings for broader audiences. Nigeria's approach with regular evening calls between technical teams and policy task forces enabled rapid response to emerging questions [15]. Public communication with several studies incorporating press releases, media engagement, and public-facing websites ensured broader dissemination of modelling insights (e.g., [34]).

**Stakeholder input.** Stakeholder input refers to the contributions, feedback, and interactions of individuals or groups with an interest in the modelling process or those who may be impacted by its outcomes. This may happen through one-time consultations, iterative collaborations or even continuous co-creation of the modelling process. Iterative collaboration may involve processes, where modellers regularly receive feedback on results to further refine their models. Co-creation goes beyond that, e.g., through joint discussions about the models throughout the process.

15 out of 33 studies benefited from stakeholder input (Table 5). In 10 of these studies, stakeholders were involved via collaborative processes, whereas the information provided in the articles did not always allow for a clear classification into iterative collaboration or co-creation. Input was mostly provided via implementers, including policy makers and clinicians, and external non-modelling experts, such as socioeconomic experts and public health professionals. Only the Pandemic Influenza Outbreak Research Modelling project in Canada reported collaboration with an affected population, in this case the local indigenous community [24]. In general, the input was incorporated in various aspects of the process such as during the selection of variables, setting assumptions, and identifying critical questions to address in the models [17,29,33,40,43].

Collaborative efforts are exemplified in New Zealand's two-round modelling approaches that allowed for clarification between teams and 'scenario sandpit' discussions [41], where policymakers requested specific simulations, and the formation of specialized working groups in Nigeria [15]. Stakeholder engagement occurred through various structured feedback mechanisms like surveys and Plan-Do-Study-Act cycles implemented among clinicians, hospital staff, commissioners, and public health consultants by National Health Service teams [38,39]. Direct communication channels with advisory groups and the media were established in multiple projects, including those in Canada with indigenous communities

**Table 5. Stakeholder input can be included through different processes and by various types of stakeholders.**

| Article | Process of stakeholder input | Type of stakeholder(s) |
|---|---|---|
| George et al., 2023 [39] | One-time consultation (at the end) | Implementers |
| Irvine et al., 2021 [38] | Iterative collaboration | Implementers |
| Shea et al., 2023 [33] | One-time consultation (in between) | Research coordination facilitator |
| Martin et al., 2020 [23] | Co-creation | Implementers |
| Tariq et al., 2021 [24] | Iterative collaboration | Affected population |
| Warde et al., 2021 [29] | Iterative collaboration | Implementers |
| Locey et al., 2020 [44] | Co-creation/ iterative collaboration | Implementers |
| Loo et al., 2024 [19] | Co-creation/ iterative collaboration | Implementers |
| McCaw and Plank, 2023 [45] | Iterative collaboration | External non-modelling experts, press |
| Howerton et al., 2023 [40] | One-time consultation (in the beginning of every modelling round) | External non-modelling experts, implementers |
| Keeling et al., 2022 [43] | One-time consultation (in the beginning) | External non-modelling experts |
| Hendy et al., 2021 [41] | Iterative collaboration | External non-modelling experts, implementer |
| Adib et al., 2021 [17] | One-time consultation (in the beginning) | External non-modelling experts |
| Aguas et al., 2020 [18] | Co-creation | External non-modelling experts |
| Abubakar et al., 2021 [15] | Co-creation/ iterative collaboration | External non-modelling experts |

The information in the articles did not always make clear whether the process of stakeholder input was one of iterative collaboration or co-creation.

through 'Community of Practice' models [24]. Additionally, international collaborations like the CoMo Consortium emphasized the role of local experts in guiding policy decisions and ensuring models reflected local contexts [18].

**Validity assessments of model results.** Modellers tried to ensure the validity of their models and results through diverse quantitative and qualitative methods across studies. Quantitative model validation approaches included comparing forecasts against actual data (e.g., [20]), calibrating models on actual data (e.g., [37]) and conducting sensitivity analyses (e.g., [34]). Cross-validation between different modelling approaches was also observed (e.g., [29]) and some studies utilized ensemble approaches, benefiting from the fact that combined forecasts often outperformed individual models (e.g., [21]). Decision-makers themselves (in North Carolina, USA) also appreciated comparisons between different models and teams, gaining confidence in the results if they confirmed each other and uncertainty if they diverged [16].

Validation frequently involved qualitative iterative processes, such as real-time assessment of forecast performance [34], allowing for continuous refinement of models. Several initiatives also implemented peer review mechanisms, exemplified by New Zealand's rapid multi-disciplinary review process [41] and the use of multiple independent modelling groups (e.g., [43]).

**Facilitators and barriers.** The included publications reflected on various facilitators and barriers to informing policies and decisions. They can be roughly clustered around the themes 'data infrastructure', 'institutional and collaborative structures', 'expertise and capacity', and 'communication and trust' (Table 6).

The modelling process was strongly facilitated by well-established institutional and collaborative frameworks. Multi-stakeholder partnerships involving experts from government, decision-making bodies, and academia led to improved predictive accuracy and efficient process execution according to, e.g., the US FluSight Network organized by the CDC [19,21]. This pre-existing Influenza modelling network also laid the groundwork for the COVID-19 and Influenza Scenario Modelling Hub, which offered long-term operational support to inform outbreak response policy in the US [19,33]. Substantial funding through research grants from the CDC further facilitated these modelling initiatives.

In contrast, inadequate institutional and collaborative structures were identified as significant barriers to the modelling process and the translation of evidence into policy. For example in Nigeria, several systematic challenges were observed, including fragmented decision-making structures, and limited availability of public health funding to support modelling efforts [15].

**Table 6. Reported facilitators and barriers for informing policies and decisions with infectious disease modelling.**

| Theme | Facilitators | Barriers |
|---|---|---|
| Data infrastructure | • Formal data-sharing agreements (e.g., [36])<br>• Public data availability (e.g., [42]) | • Quality issues and inconsistencies (e.g., [37])<br>• Reporting delays (e.g., [22])<br>• Integration challenges (e.g., [25])<br>• Limited access to granular data (e.g., [20]) |
| Institutional and collaborative structures | • Pre-existing networks (e.g., [33])<br>• Multi-stakeholder partnerships (e.g., [19])<br>• Funding (e.g., [19,24]) | • Fragmented decision-making (e.g., [15])<br>• Limited funding (e.g., [15]) |
| Expertise and capacity | • Diverse expertise (e.g., [15])<br>• Professionals in decision making bodies to interpret models and their results (e.g., [16]) | • Limited national modelling capacity (e.g., [17])<br>• Knowledge gaps (e.g., [22])<br>• Insufficient consideration of human behaviour in models (e.g., [16]) |
| Communication and trust | • Participatory approaches (e.g., [18])<br>• Feedback loops (e.g., [18])<br>• Active engagement platforms (e.g., [19])<br>• Good visualisations (e.g., [16]) | • Challenges in conveying uncertainty (e.g., [43])<br>• Misinterpretation of results (e.g., [45])<br>• Trust building challenges (e.g., [17])<br>• Reliance on key individuals (e.g., [18])<br>• Confusion about differing results from multiple models (e.g., [16]) |

Note that we include examples for the individual facilitators and barriers where the facilitator/barrier is clearly highlighted. This does not exclude that other included articles also consider the respective facilitator/barrier.

13 of the included studies highlighted barriers in data infrastructure [20,22,25,34,46]. In general, the study of decision makers in North-Carolina mentions hesitancies in the use of modelling due to "concerns about the quality of data used in models" [16]. Delayed and inconsistent epidemiological data reporting emerges as the most pervasive challenge across modelling efforts [20,22,25,28,34,37,39,46]. A lack of granularity of data hindered addressing subpopulation-specific questions [16,25,42]. Sherratt et al. show that "biased representations of subpopulations in each data source" cause differences in transmission estimates, highlighting the need for decision-makers to be made aware of source populations of used data [42]. The examples, where detailed data was available, were in turn able to provide more nuanced results [20,42,43]. Access to data was sometimes enabled through partnerships: collaboration with field staff in Bangladesh [22], with ministries in Austria [28], United Kingdom [43] and Costa Rica [26], and other policy makers [18]. However, the use of modelling was also highlighted as a tool to correct for bad data quality [18,20,39], e.g., through hierarchical modelling to account for missing data [20] and by modelling data delays [22]. Another key type of data that was mentioned to be lacking was behavioral data [16,43]. It made it difficult to accurately model human behavior. An extended description of data-related challenges is described by Steinberg et al. [25].

6 of the reviewed articles highlighted uncertainties in the projections due to the novel nature of SARS-CoV-2 and the use of varying parameters and assumptions by different modelling teams. This led to a discordant set of results presented to decision-makers, fostering confusion, mistrust, and diminishing interest in the models among decision-makers, as illustrated by a study in the Eastern Mediterranean region [17]. Keeling et al. also noted the challenge of effectively communicating such uncertainties in long-term projections to policymakers and the public [43]. The results were frequently misinterpreted as definitive predictions in the media and decision-making contexts, which, in some cases, led to the potential dismissal of the modelling outcomes [45].

Local decision-makers were especially concerned about translating model results based on national data and perspectives into their local context and expressed scepticism towards the ability of models to appropriately incorporate human behaviour [16]. In their case, having internal professionals to interpret and translate the results was reported as helpful.

## Discussion

In this scoping review, we aimed to get an overview of the infectious disease modelling literature that assesses its utility in informing decisions for outbreak response. We included 33 peer-reviewed articles, none of which systematically assessed

the utility of their modelling effort for decisions. Instead, authors more subjectively reflected on their modelling effort(s) and highlighted key aspects they deemed relevant for utility. Hence, this review does not present objective utility assessments but summarises those subjective reflections. We focused on different aspects of the described modelling-for-decisions pathways: the context of the outbreaks, whether the authors and initiatives stemmed from the modelling or policy field, the objectives of the policy and the role of modelling in achieving these objectives, the types of decision-makers, how modelling informed decisions, and the facilitators and barriers in the pathway.

Our analysis reveals that there are multiple processes through which modelling is able to inform decisions. A key finding is that modellers perceive the modelling-for-decisions pathway to be enhanced by well-established institutional structures and collaborative frameworks, particularly through multi-stakeholder partnerships between policymakers and researchers [15,19,33,36,43]. These frameworks not only facilitated the rapid generation of scientific evidence but also promoted open communication and trust between decision-makers and modellers, which was critical for the co-creation of decision-relevant evidence during the pandemic.

This is in line with general findings about policy that is informed by any kind of scientific evidence. To systematically evaluate evidence-informed policymaking, various models and frameworks have been developed to organize existing knowledge in policy development [47]. These frameworks, commonly utilized in health policy and systems research, include the Ottawa Model of Research Use (OMRU) [48], The Framework for Research Dissemination and Utilization (FRDU) [49] and The Canadian Health Services Research Foundation model (CHSRF) [50]. Many of the facilitators of the modelling-for-decisions pathway identified in this review closely align with the guiding principles of those frameworks for evidence-informed policy making. For example, both the OMRU and FRDU frameworks emphasize the importance of involving multiple stakeholders at various levels of the healthcare system in the knowledge-to-policy translation, while the CHSRF advocates for the creation of communication channels among researchers, decision makers, and research funders to foster trust and facilitate the mutual exchange of knowledge. Such collaborative approaches, particularly for modelling-for-decisions pathways, are also praised by others [5,51,52], which further highlights the importance of bringing scientists and decision makers together for modelling initiatives. In total, this emphasises the partial transferability of general evidence-informed policy research to modelling-evidence-informed policy research.

Our review also identified several barriers that were thought to be hindering the utility of modelling for the decision-making process. The barriers to using infectious disease modelling in policymaking emerged across several key themes. Data challenges—unavailable, unreliable, or delayed—undermined the accuracy of models, while integration issues tangled the flow of information. Institutional hurdles, like fragmented decision-making, stifled progress, especially where limited funding further constrained efforts [15]. Limited expertise and national modelling capacity, compounded with knowledge gaps regarding the disease, further deepened these setbacks. Communication between modellers and decision-makers proved to be another stumbling block in the modelling-for-decisions pathway as uncertainties in projections sowed confusion and eroded trust among decision-makers [53]. Misinterpretation of modelling results by the media only amplified this uncertainty and diminished confidence in the models, undermining their potential impact on policy decisions [45].

In this review, we presented the facilitators, barriers and all other characteristics of modelling-for-decisions pathways independent of each other and were not able to identify relationships and dependencies between those characteristics. This can be explained by the diversity of existing modelling-for-decisions pathways and experiences; the available evidence and level of detail gathered from a scoping review may not be large enough to allow for a more detailed analysis.

## Limitations

A limitation of our study is its eligibility criteria: To use the 'remove duplication function' of Ovid and thereby ensure a manageable scope of the review, we had to limit the number of retrieved articles in the search to below 6000. We achieved this by adapting our eligibility and search criteria. This involved focusing on only key pathogens and the exclusion of studies on cost-effectiveness and pharmaceutical interventions, studies looking only at single non-pharmaceutical

interventions for COVID-19, as well as not-peer-reviewed publications. Yet, we acknowledge that in doing so, we may have created bias in our sample of papers and overlooked some valuable insights. For example, the substantial body of work from modellers reporting specific non-pharmaceutical interventions for COVID-19 may represent a different experience of the modelling-for-decisions pathway or emphasise different facilitators and barriers.

Another subset of literature that we have not actively considered is on machine learning-based approaches to inform decisions. Although some of the included literature considers machine learning-based models, such as Bayesian inference models or even neural network-based models, our search strategy and choice of databases aimed to find literature that considers more explicit modelling approaches. However, it is important to note that more black-box-type machine learning approaches are seen as promising directions in infectious disease modelling and public health policy [54,55].

Further, our review is likely affected by publication biases: Some of them are related to the fact that the included articles are subjective assessments, mostly reflecting the perspectives of the authors of modelling studies on what they considered important for the utility of the process. Hence, this review can only capture what authors were willing to share. Authors may prefer not to report or only selectively describe attempts to inform decisions that were perceived as particularly challenging or unsuccessful. In particular, these challenges may involve interpersonal or political tensions, or institutional power dynamics. This might also mean that only more manageable barriers were represented in our analysis. Additionally, fully understanding modelling-for-decisions pathways requires more perspectives of decision makers and other stakeholders.

We also acknowledge that much of the modelling that shapes real world decision making eludes the academic record. This includes grey literature such as situational reporting, software tools, and other real-time outputs that may be too fluid for traditional academic reporting structures [56,57].

While a future systematic review could omit some of our restrictive criteria to capture a broader picture of the existing literature, we believe our approach has nevertheless captured key characteristics of the evidence body because other commentaries and viewpoints on the same topic support our own findings [1–3,5,6,51,52,58–61].

Owek et al. have conducted a multi-method study on this topic, involving interviews, workshops, mapping exercises and a scoping review. Their review concentrated on COVID-19 in LMICs and used a different search strategy, including different search terms [10]. Although we did not actively exclude LMIC settings from our analysis, more papers from HICs met our inclusion criteria. To ensure a more diverse overview, we therefore also included articles from Owek et al.'s scoping review if they matched our eligibility criteria. While these described modelling efforts informing LMIC policy makers, they mainly involved HIC modellers [17,18]. This points to a possible lack of local modelling expertise used in LMIC policy and also means that modelling efforts involving LMIC modellers are underrepresented in our review. However, the comparison between our findings and those of Owek et al.'s focusing on LMICs shows many similar insights, emphasizing the robustness of our results [10]: Both reviews highlight that the efficacy of modelling-for-decision efforts seems to increase with pre-existing connections between modellers and policy makers, as well as open communication and trust between them. A key difference is that Owek et al. reported rivalry among modellers and officials' dislike of research in LMICs as barriers to effective knowledge translation [10]. These barriers were not found in our review and may have been identified from the other activities of their multi-method study.

## Conclusion and outlook

In this scoping review, we investigated the role of mathematical modelling in shaping decisions during infectious disease outbreaks, and identified the factors that practitioners considered crucial for the effectiveness of this process. The initial objective of this review was to identify literature evaluating the utility of mathematical modelling in informing decisions during infectious disease outbreaks, which could inform a future systematic review. However, we did not find any studies that provided systematic assessments of the utility or evaluations of the pathways for modelling-informed decisions. This also contributed to the fact we were not able to find clear relationships between characteristics of the pathway. As

we mostly found reflections by modellers, we may have missed important perspectives of decision makers and other stakeholders. While a more diverse authorship of this review may have influenced the study protocol and interpretation of results, it cannot make up for this lack of perspectives in the literature.

Together with other previous commentaries (e.g., [5]), these issues highlight the need for tools and studies that systematically perform assessments and evaluations of modelling-for-decision pathways. These evaluations could involve prospectively defining criteria in collaboration with decision-makers that would enable retrospective evaluation once the modelling work has completed, but also more general reflections of modellers on model accuracy, utility and the relationship between the two [62]. Modelling work to inform decisions has the potential to do harm as well as save lives, and only through honest evaluation of past efforts will it be possible to come to general insights on how to maximise its positive impact on human health.

## Supporting information

**S1 PRISMA Checklist.  Checklist for the PRISMA Scoping Review guidelines.**
(PDF)

**S1 File.  Document containing supplementary figures and tables.** S1 File Fig A. Screenshot of final search strategy and query on Ovid. S1 File Table A. Rationales for the search strategy and query on Ovid. S1 File Fig B. Number of included articles by year of publication. S1 File Fig C. Number of included articles by considered target population in the model. S1 File Fig D. Number of included articles by considered disease or pathogen in the model.
(PDF)

**S1 Data.  Excel file containing the extracted data.**
(S1_Data.XLSX)

## Acknowledgments

We thank Francisco Pozo Martin for his guidance on conducting a literature review. We are grateful to Liza Hadley and Paula Christen for their feedback on our results. Lastly, we thank the reviewers for their constructive comments and suggestions, which have improved this review substantially.

## Author contributions

**Conceptualization:** Emil Nafis Iftekhar, Sophie Alice Müller, Julia Fitzner, Johanna Hanefeld, Sebastian Funk.

**Data curation:** Duaa Rao, Aleena Tanveer, Emil Nafis Iftekhar, Sophie Alice Müller, Katharine Sherratt, Katharina Heldt.

**Formal analysis:** Duaa Rao, Emil Nafis Iftekhar.

**Funding acquisition:** Julia Fitzner, Johanna Hanefeld, Sebastian Funk.

**Investigation:** Duaa Rao, Aleena Tanveer, Emil Nafis Iftekhar.

**Methodology:** Duaa Rao, Aleena Tanveer, Emil Nafis Iftekhar, Sophie Alice Müller, Katharine Sherratt, Paola Carrillo-Bustamante, Sebastian Funk.

**Project administration:** Duaa Rao, Aleena Tanveer, Emil Nafis Iftekhar, Sophie Alice Müller.

**Supervision:** Emil Nafis Iftekhar, Katharine Sherratt, Julia Fitzner, Johanna Hanefeld, Sebastian Funk.

**Validation:** Duaa Rao, Aleena Tanveer, Emil Nafis Iftekhar, Katharine Sherratt, Sebastian Funk.

**Visualization:** Duaa Rao, Aleena Tanveer, Emil Nafis Iftekhar.

**Writing – original draft:** Duaa Rao, Aleena Tanveer, Emil Nafis Iftekhar.

**Writing – review & editing:** Duaa Rao, Aleena Tanveer, Emil Nafis Iftekhar, Sophie Alice Müller, Katharine Sherratt, Klara Röbl, Julia Fitzner, Sebastian Funk.

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
