## [Decision Letter · Decision Letter 0]

3 Jun 2025

PGPH-D-25-00570

The utility of infectious disease modelling in informing policy for outbreak response: a scoping review

Dear Dr. Iftekhar,

Thank you for submitting your manuscript to PLOS Global Public Health. After careful consideration, we feel that it has merit but does not fully meet PLOS Global Public Health’s publication criteria as it currently stands. Therefore, we invite you to submit a revised version of the manuscript that addresses the points raised during the review process.

We look forward to receiving your revised manuscript.

Kind regards,

Charin Modchang, Ph.D.

Academic Editor

Journal Requirements:

Additional Editor Comments (if provided):

We have received comments from the reviewers. Please address them accordingly. Thank you.

Reviewers' comments:

Reviewer's Responses to Questions

**Comments to the Author**

1. Does this manuscript meet PLOS Global Public Health’s publication criteria ? Is the manuscript technically sound, and do the data support the conclusions? The manuscript must describe methodologically and ethically rigorous research with conclusions that are appropriately drawn based on the data presented.

Reviewer #1: Yes

Reviewer #2: Yes

Reviewer #3: Yes

2. Has the statistical analysis been performed appropriately and rigorously?

Reviewer #1: Yes

Reviewer #2: N/A

Reviewer #3: N/A

3. Have the authors made all data underlying the findings in their manuscript fully available (please refer to the Data Availability Statement at the start of the manuscript PDF file)?

Reviewer #1: Yes

Reviewer #2: Yes

Reviewer #3: Yes

4. Is the manuscript presented in an intelligible fashion and written in standard English?

Reviewer #1: Yes

Reviewer #2: Yes

Reviewer #3: Yes

5. Review Comments to the Author

Reviewer #1: This is an important work that presents the use of math modelling into the policy despite its presence since decades. I have no comments - authors presented very well with strong methodology with proper interpretations and conclusion.

Reviewer #2: The author have conducted a scoping review “to assess how infectious disease modelling inform policy, focusing on challenges and facilitators of translating modelling insights into actionable policies.

1. As mentioned by the authors, this manuscript is quite similar to ref [9], and the authors attempt to distinguish findings between the two manuscripts in the discussion.

2. A point of contention in phrasing “modelling to policy” that is mentioned several times. A more correct phrasing would be modelling to decision-support as many of these models supporting resource allocation are not really geared into action that that results in a policy revision or new policy being written.

3. What is not clear from the review is the impact made… i.e. who listened to what evidence and how that was consumed?

4. Given that the majority of LMIC articles in the came from [16,17] – a consortium originated and led by Oxford University, to what extent does the review represent global experience?

5. Important themes are highlighted in the review, but they are done univariately. It would have been useful to link the barriers with other characteristics of the candidate studies. E.g. Was it mostly ABM models that considered infrastructural issues as a barrier etc. Is that possible with this sample size? This could be presented diagrammatically.

6. Given that the study did not achieve its aim of assessing the “utility of mathematical modelling in informing policy during infectious disease outbreaks”, is there a study design that renders this possible accounting for the nature of decision-making?

Reviewer #3: This manuscript addresses an important and timely topic: how infectious disease modelling informs public health policy during outbreaks. The review is methodologically sound, well-structured, and offers valuable insights into facilitators and barriers along the modelling-to-policy pathway. However, several areas could be clarified or expanded to strengthen the manuscript’s utility for both researchers and decision-makers.

Major Points

1. Definition of Mathematical Modelling

The manuscript would benefit from a clearer definition of "mathematical modelling," especially for readers outside the field. The current discussion leans heavily on traditional infectious disease models (e.g., differential equations, agent-based, stochastic models), but increasingly, governments and stakeholders are engaging with machine learning (ML)-based approaches. While neural networks are briefly mentioned, there is no discussion of how these fit within the modelling-to-policy context. Additionally, much of the ML-related literature is not indexed in the databases searched (e.g., dblp.org, IEEE, NeurIPS, ICML), and this limitation should be noted—particularly as these models are now being shared with and used by decision-makers. This doesn’t require changing the scope of the review but should be clarified in the framing and limitations.

2. Clarification of Owek et al. [9]

The review refers to Owek et al. as a scoping review, but the study also involved interviews, workshops, and mapping exercises. These methods enabled the authors to explore interpersonal, political, and structural barriers in greater depth. Clarifying the methodological differences will help readers understand why certain issues (e.g., rivalry between modellers and resistance from officials) appeared in their findings but not in this review.

3. Limits of Self-Reported Reflections

The review should more clearly acknowledge that most findings are based on reflections from modellers about their own work. As such, interpersonal or political tensions, institutional power dynamics, or failed engagements may not be reported or may be selectively described. While the review notes this in passing, more explicit discussion of what cannot be captured through this method would improve the interpretation of results. This should be reflected both in the Limitations and in the framing of the Results and Discussion sections.

4. Modelling Choices and Policy Uptake

The review does a good job categorizing model types, but it misses the opportunity to explore how the choice of model influenced clarity, trust, or uptake. For example, ensemble models (e.g., FluSight Network) increased credibility when projections aligned, while more complex models may have been harder to communicate when data was limited. A comparative analysis—even if qualitative—of how deterministic vs. stochastic, statistical vs. mechanistic, or ML vs. traditional frameworks influenced accessibility, interpretability, or actionability would be a valuable addition.

5. Data Quality and Availability

Data-related challenges are mentioned but discussed in general terms. Given how critical this issue is, a deeper discussion would benefit both modelers and stakeholders. Consider expanding on common data limitations (e.g., lack of disaggregated or behavioral data, legal constraints, reporting delays) and how these impacted model design or policy relevance. This would offer practical insights for those working to improve modelling processes and data systems.

6. Stakeholder Involvement

The review notes stakeholder engagement in 15 studies but doesn't break down the depth or phase of this involvement. Were stakeholders involved in framing the initial questions, selecting interventions, or only reviewing final outputs? Were they funders, implementers, or external reviewers? A clearer typology (e.g., co-creation, iterative collaboration, one-time consultation) would help readers understand how engagement influenced policy relevance and uptake.

7. COVID-19 vs. Non-COVID-19 Comparison

Given that most studies focused on COVID-19, it would be helpful to compare these to the non-COVID-19 studies to assess whether modelling uptake, stakeholder dynamics, or data issues differed. Policymaker expectations and urgency were different during the pandemic, which may have shaped how modelling was used.

8. Inclusion of Stakeholders in the Review Itself

Consider discussing whether stakeholders (e.g., policy decision-makers, public health authorities) were involved in designing or interpreting this review. Including such voices might have added valuable perspectives on what makes modelling useful in practice.

Minor changes:

The word mpox should be lower case if not at the start of a sentence

Check that all citations are incorporated into endnote ({Finger, 2019 #4019})

“modelling-to-policy pathway” is used throughout but once appears as “modelling to policy process”

6. PLOS authors have the option to publish the peer review history of their article (what does this mean? ). If published, this will include your full peer review and any attached files.

**Do you want your identity to be public for this peer review?** For information about this choice, including consent withdrawal, please see our Privacy Policy .

Reviewer #1: No

Reviewer #2: No

Reviewer #3: No

---

## [Decision Letter · Decision Letter 1]

13 Aug 2025

The utility of infectious disease modelling in informing decisions for outbreak response: a scoping review

PGPH-D-25-00570R1

Dear Iftekhar,

We are pleased to inform you that your manuscript 'The utility of infectious disease modelling in informing decisions for outbreak response: a scoping review' has been provisionally accepted for publication in PLOS Global Public Health.

Best regards,

Charin Modchang, Ph.D.

Academic Editor

Thank you for addressing all of the reviewers' comments.

Reviewer Comments (if any, and for reference):

Reviewer's Responses to Questions

**Comments to the Author**

1. If the authors have adequately addressed your comments raised in a previous round of review and you feel that this manuscript is now acceptable for publication, you may indicate that here to bypass the “Comments to the Author” section, enter your conflict of interest statement in the “Confidential to Editor” section, and submit your "Accept" recommendation.

Reviewer #2: All comments have been addressed

Reviewer #3: All comments have been addressed

2. Does this manuscript meet PLOS Global Public Health’s publication criteria ? Is the manuscript technically sound, and do the data support the conclusions? The manuscript must describe methodologically and ethically rigorous research with conclusions that are appropriately drawn based on the data presented.

Reviewer #2: Yes

Reviewer #3: Yes

3. Has the statistical analysis been performed appropriately and rigorously?

Reviewer #2: Yes

Reviewer #3: N/A

4. Have the authors made all data underlying the findings in their manuscript fully available (please refer to the Data Availability Statement at the start of the manuscript PDF file)?

Reviewer #2: Yes

Reviewer #3: Yes

5. Is the manuscript presented in an intelligible fashion and written in standard English?

Reviewer #2: Yes

Reviewer #3: Yes

6. Review Comments to the Author

Reviewer #2: One minor point for the authors to clarify on the manuscript is if the CoMo consortium has more HIC modellers than LMIC modellers? The point I made was that the models were developed in one HIC institution. The authors should be clear before committing to that statement.

Reviewer #3: Thank you for so diligently and kindly responding to each of the points raised in the submitted review. All points have been addressed. Thank you.

7. PLOS authors have the option to publish the peer review history of their article (what does this mean? ). If published, this will include your full peer review and any attached files.

**Do you want your identity to be public for this peer review?** For information about this choice, including consent withdrawal, please see our Privacy Policy .

Reviewer #2: No

Reviewer #3: No
